# Nanoparticle Uptake in the Aging and Oncogenic *Drosophila* Midgut Measured with Surface-Enhanced Raman Spectroscopy

**DOI:** 10.3390/cells13161344

**Published:** 2024-08-13

**Authors:** Maria Christou, Ayobami Fidelix, Yiorgos Apidianakis, Chrysafis Andreou

**Affiliations:** 1Department of Biological Sciences, University of Cyprus, Nicosia 2109, Cyprus; 2Department of Electrical and Computer Engineering, University of Cyprus, Nicosia 2109, Cyprus

**Keywords:** *Drosophila melanogaster*, surface-enhanced Raman scattering, SERS, gut disease, nanoparticle uptake

## Abstract

Colorectal cancer remains a major global health concern. Colonoscopy, the gold-standard colorectal cancer diagnostic, relies on the visual detection of lesions and necessitates invasive biopsies for confirmation. Alternative diagnostic methods, based on nanomedicine, can facilitate early detection of malignancies. Here, we examine the uptake of surface-enhanced Raman scattering nanoparticles (SERS NPs) as a marker for intestinal tumor detection and imaging using an established *Drosophila melanogaster* model for gut disease. Young and old Oregon-R and *w^1118^* flies were orally administered SERS NPs and scanned without and upon gut lumen clearance to assess nanoparticle retention as a function of aging. Neither young nor old flies showed significant NP retention in their body after gut lumen clearance. Moreover, tumorigenic flies of the *esg-Gal4/UAS-Ras^V12^* genotype were tested for SERS NP retention 2, 4 and 6 days after Ras^V12^ oncogene induction in their midgut progenitor cells. Tumorigenic flies showed a statistically significant NP retention signal at 2 days, well before midgut epithelium impairment. The signal was then visualized in scans of dissected guts revealing areas of NP uptake in the posterior midgut region of high stem cell activity.

## 1. Introduction

Colorectal cancer (CRC) is the third most diagnosed malignancy worldwide in men and women. According to GLOBOCAN, the age-standardized CRC worldwide incidence rate in 2022 was 21.9 in men and 15.2 in women, per 100,000 people. However, in people younger than 50, rates have been increasing by 1–2% a year since the mid-1990s [1]. Colonoscopy with biopsy acquisition is the gold standard for the identification of CRC lesions. This method presents a crucial economic burden on the healthcare system, as well as discomfort to the patient along with risk of complications, such as bleeding [2]. Moreover, colonoscopy relies on the visual assessment of lesions, causing early-stage lesions to be missed [3]. More accurate and less invasive diagnostic methods can improve patient compliance and early detection.

Raman spectroscopy with surface-enhanced Raman scattering nanoparticles (SERS NPs) as contrast agents is actively explored as a method for the early detection of cancer. SERS is a surface-sensitive optical technique that enhances Raman scattering by molecules near metal nanoparticles (commonly gold or silver), leading to a significant amplification of the Raman spectrum intensity [4]. This optical technique was used as a highly accurate and non-destructive tool in the imaging and diagnosis of cancer in preclinical studies [5]. Raman endoscopy has been demonstrated in rodents [6] and larger animals [7,8], where SERS nanoprobes were shown to provide higher detection sensitivity and molecular information related to the lesions. Recently, orally administered SERS NPs have been reported to reveal inflammatory bowel disease in mice [9]. Additionally, SERS was used for liquid biopsies of biofluids and breath, capturing the metabolic fingerprint associated with lung, breast, colorectal, ovarian, and oral cancer while also enabling a differential diagnosis between cancer types [10,11]. While most studies focus on the clinical translation of the technique, a more fundamental understanding of the interactions of SERS nanoparticles with the tumorigenic intestinal epithelium may uncover new routes for diagnosis and drug delivery. This can be achieved by using facile and genetically amenable model organisms that share specific features of human disease.

*Drosophila melanogaster*, the common fruit fly, has been increasingly used as a model for human gut disease [12]. Its genome contains fewer copies per gene compared to humans, and about 65% of the genes responsible for human diseases have homologs in flies [13]. It is well suited for the study of intestinal stem cell physiology during aging, stress, and infection, and for modeling human intestinal malignancies [12], especially because signaling pathways responsible for tissue invasion and growth of cells in mammals have analogous functions in flies [14]. Accordingly, multigenic models of human colon cancer can be generated based on the cancer genome Atlas of patients’ data [15]. Moreover, intestinal microbes can be studied in *Drosophila* to elucidate mechanisms underlying host-gut microbiota interactions [16]. These features, together with the rapid generation time, low maintenance costs, and the availability of powerful genetic tools, make *Drosophila* a powerful model in the study of complex diseases, including cancer [17,18,19].

In this brief report, we explored orally administered SERS NPs as a marker for diseased gut in young, old, and tumorigenic flies. We performed Raman measurements of the abdomen of flies fed with SERS NPs and corresponding controls. Additionally, for the tumorigenic flies, excised midguts were imaged to identify the localization of the SERS NPs within them. 

## 2. Materials and Methods

-Fly production

Oregon-R, *w^1118^*, *esg-Gal4 UAS-GFP tub-Gal80^ts^* and *UAS-Ras^V12^* (BDSC#64196) were maintained in bottles containing 1.2% agar 6% cornmeal, 3% yeast, 5% sucrose, supplemented with the food preservatives Tegosept and propionic acid. To study the effect of oncogene expression in the *Drosophila* midgut progenitor cells, *esg-Gal4 UAS-GFP tub-Gal80^ts^* were crossed to *UAS-Ras^V12^* and offspring (*esg^ts^-Ras^V12^*) were collected for four days at 18 °C, then shifted to 29 °C to activate the oncogene for 2, 4, or 6 days [20].

-SERS NP synthesis

The SERS nanoparticles were synthesized based on existing protocols [21]. Chemicals were purchased from Sigma Aldrich and used without further purification. Briefly, for the synthesis of gold nanostar cores, an ascorbic acid solution comprising 8.45 g of ascorbic acid in 800 mL of 4 °C deionized water (DI) was mixed using a magnetic stirrer to induce a vortex; rapidly, 8 mL of 20 mM tetrachloroauric acid (HAuCl4) solution also at 4 °C were added, resulting in the rapid formation of nanostars and a transformation of the solution to a dark blue color. The nanostar suspension was then carefully transferred into 50 mL conical tubes and subjected to centrifugation for 20 min at 4 °C and 3200× *g*. Following centrifugation, the supernatant in each tube was aspirated, leaving approximately 200 μL of the solution. The solution in each tube was agitated using a micropipette, ensuring proper suspension of the nanoparticles before collection. Subsequently, the nanoparticle suspension underwent dialysis in a cassette (MWCO: 3500), against 2 L of 4 °C DI water, for three days, with daily water changes to facilitate purification and further refinement of the synthesized gold nanostar cores.

The formation of the silica shell around the gold nanostar cores involved the preparation of two distinct tubes of reagents. In a 50 mL plastic tube, a mixture comprising 10 mL isopropanol, 500 μL tetraethyl orthosilicate (TEOS), 200 μL deionized water (DI), and 2 μL of 20 mM of the dye IR780 iodide was subjected to a steady vortex. In a 15 mL tube, a mixture of 3 mL ethanol, 200 μL ammonium hydroxide, and 1.2 mL of the previously synthesized nanostars was prepared. The contents of the smaller tube were rapidly introduced into the vortex of the larger tube, initiating a 15 min incubation at room temperature. Post-incubation, the reaction was quenched by adding ethanol to reach a volume of 50 mL. Subsequent to quenching, the solution underwent centrifugation for 20 min at 3200× *g* and 4 °C. The supernatant was then carefully aspirated, leaving approximately 0.5 mL of the solution, containing the silica-coated SERS NPs. To resuspend the nanoparticles, 1 mL of ethanol was added, and the resulting solution was transferred to a 1.5 mL centrifuge tube. A series of washing steps followed, involving three rounds of centrifugation at 11,000× *g* for 4 min, aspirating the supernatant, and resuspending the pellet through ultrasonication for 1 s, each time using ethanol. An additional washing step replaced ethanol with DI water. Finally, during the last resuspension of the pellet, 1 mL of DI water was added, concluding the process and allowing for the storage of the SERS NPs at 4 °C.

-Administration of SERS nanoparticles

The SERS NPs were administered to female flies by feeding. To synchronize feeding, all flies underwent a 3 h starvation period in empty fly vials. Subsequently, the flies were transferred to a 50 mL tube equipped with 0.9 mm holes for aeration; the tube’s lid, featuring 1.2 mm holes, had Whatman paper placed on it. To initiate the feeding process, 200 μL of SERS NP suspension, prepared by diluting the silicated gold nanostars in a 4% sucrose solution at a 4× dilution of the master mixture, was carefully transferred onto the Whatman paper. The lid was then securely encased with parafilm to prevent evaporation. The flies were allowed to feed for 24 h at a controlled temperature of 25 °C. 

This study was designed with two arms: the flies were subjected to Raman measurements immediately after the 24 h incubation period, or they were allowed to clear their gut post-feeding. To allow clearance, the tube’s lid was replaced with cotton, and the flies remained for an additional 3 h period without access to any food. Before scanning, the tubes were parafilm wrapped and transferred to ice to anesthetize the flies and Raman measurements were performed one fly at a time. 

-Whole-fly Raman measurements

Raman measurements were taken with a 785 nm diode laser (Ramulaser^TM^, SterllarNet Inc., Tampa, FL, USA) at 240 mW output power and 210 μm in spot diameter. For the fly abdomen measurements, the process commenced with the transfer of a 5 μL drop of nanoparticles to a microscope slide, to allow for laser focusing and provide a positive control (reference) spectrum. Subsequently, anesthetized flies, taken one at a time, were carefully positioned on the slide, with the laser directed at their abdomen. Each fly measurement was performed with 1 s time integration with five repetitions to enhance accuracy. Post-measurement, every result underwent processing steps in Matlab (Mathworks, Natick, MA, USA) version 2023b, using the PLS Toolbox version 9.2.1 (EigenVector Reasearch Inc., Manson, WA, USA), including dark subtraction, baseline adjustment (Whittaker filter set at 100), and smoothing (Savoy-Golay filter, set at 20), to refine and standardize the acquired data.

-Midgut preparation for Raman imaging

Midguts of *esg-Gal4/UAS-Ras^V12^* females were dissected to allow examination of the nanoparticle distribution within the tumorigenic gut. Following 24 h of feeding with SERS NPs, the flies underwent a 3 h starvation period at 25 °C. Subsequently, the flies were anesthetized using CO_2_ and transferred onto ice to maintain their state of unconsciousness. Flies were dissected under a stereoscope and midguts were placed in a drop of 1 × PBS solution, within a 30 min timeframe. For fixation, midguts were transferred to glass wells containing 300 μL of a 4% formaldehyde solution for 30 min. Following formaldehyde treatment, three washing steps were carried out using 300 μL of 1 × PBS each time. The fixed guts were then carefully transferred and aligned onto a glass slide containing 20 μL of 1 × PBS solution. Coverslips were placed over the midguts to stabilize them, followed by the addition of nail polish around the coverslip edges. The prepared gut specimens were then subjected to Raman scanning, allowing for the evaluation of the distribution of SERS NPs in and around the midguts. Midguts were imaged similarly to whole flies. A 5 μL drop of nanoparticles was transferred to a slide for laser adjustment. The area containing the guts was then measured on both the x and y axes, with specific steps chosen on each axis (stepsize set at 200 µm). Following the customization of appropriate settings, an area scan was performed for each set of midguts using the same Raman 785 nm spectrometer as for the whole-fly measurements. The spectral maps obtained from the dissected guts underwent processing, including dark subtraction, baseline adjustment (set at 100), and smoothing (Savoy-Golay filter, set at 20). 

-Smurf assay for gut leakiness

For the Smurf assay, 0.5% Bromophenol Blue, 4% sucrose in ddH_2_O was vigorously shaken and pH was adjusted to 7 with a small volume of acid or base. Sterilized cotton balls were placed at the bottom of empty glass vials, and 5 mL of the blue dye solution was added to each vial. Female flies were starved for 5 h in empty vials at 25 °C, transferred into the vials, and the vials were sealed with sterilized cotton. The flies were fed for 24 h at 25 °C, and categorized under a stereoscope as Smurfs (diffused body blue) or non-Smurfs (intestine restricted blue).

-Statistical analysis

Data analysis and visualization were performed in Matlab (Mathworks). Student’s two-sample *t*-test was used to establish differences in the Raman signals between nanoparticle-fed flies and controls, as it compares the means of the two populations. For the Smurf assay, Fisher’s exact test was used, as it is suitable for analysis of contingency tables, and it is suitable for small sample sizes.

## 3. Results

### 3.1. SERS Nanoparticle Characterization

Synthesized SERS nanoparticles were serially diluted to assess their Raman signal as a function of concentration. The nominal concentration for this synthesis methodology is approximately 1 nM. The characteristic Raman peaks of the reported dye, namely, the main peak at 1203 cm^−1^ and the two peaks at approximately 520 and 557 cm^−1^, were visible down to at least the 64-fold dilution (Figure 1a). The intensity of the 1203 cm^−1^ peak diminished with decreasing concentration, following a logarithmic trend, as shown in Figure 1b. Thus, the SERS nanoparticles can provide a bio-orthogonal signal allowing their detection in the fly gut, in a fairly quantitative manner.

### 3.2. Nanoparticle Uptake by the Gut of Young and Old Flies

To detect the uptake and retention of orally administered NPs in relation to the fly’s age, two *Drosophila* strains were assessed. Wild-type Oregon-R (OR) and *w^1118^* at 4 and 30 days of age were tested. First, to verify the impact of aging on the fly intestines, we assessed its permeability via the Smurf Assay [22,23]. Female flies were assessed, as they were reported to provide more relevant age-related results [24]. Fisher’s exact test was preferred as it provides an exact *p*-value even for small sample sizes. We found significantly increased permeability in the 30-day-old flies of both *Drosophila* strains (Figure 2, Table 1). Younger flies were more readily available than older flies, as they are generated faster, and were tested in higher numbers. Due to the difference in sample size, a statistical power calculation was performed with the expected rate of Smurf incidence to ensure sufficient statistical power. Specifically, considering that the Smurf assay is a dichotomous (yes/no assay) with a typical anticipated incidence of 50% for old OR flies and 10% for young OR flies and that we require a 95% confidence interval, the minimum sample size required for 0.9 statistical power is 14 old and 56 young flies. Similarly, for the *w^1118^* flies, if we consider 40% and 15% incidence for old and young flies, respectively, 0.9 power requires at least 18 old and 36 young flies. Our tested populations, shown in Table 1, were greater than these, ensuring statistical power greater than 90%.

For the Raman study, flies were fed with SERS NPs for 24 h and divided in the two study arms: one group was scanned immediately after the end of the 24 h period (“no clearance” condition) and the other group was scanned after an additional 3 h of intestinal lumen clearance. The acquired Raman spectra demonstrated low intensities and rich spectral signals for all the experimental conditions (Figure 3a). Control flies without exposure to SERS NPs demonstrated many intrinsic Raman peaks stemming from the fly’s body; these can confound direct spectral identification of the nanoparticle signature. Spectral deconvolution was performed using constrained linear regression, namely non-negative Least Squares (nn-LS). Spectra of the pure nanoparticles and of control flies without NP exposure were used as references. For each strain and age group, the corresponding control flies were scanned to provide the “background” reference, ensuring that the appropriate spectral features for each strain were accounted for, as the intrinsic Raman signals have been reported to change as the flies age [25]. Student’s two-sample *t*-test was used to compare the scores of the pairwise populations. This test is appropriate for comparing independent measurements from two distinct populations, as we have here, under the assumption that the intensities within each population are normally distributed. Young and old flies, regardless of strain or experimental condition, demonstrated tentative nanoparticle signal, when compared NP-fed with control vehicle-fed flies (Figure 3b). Aged *w^1118^* flies allowed no clearance time before the Raman scan demonstrated high SERS NP signals significantly different than the corresponding control flies. When allowed a clearance period, the flies demonstrated lower signals, that were not significantly different than the controls, indicating that the nanoparticles detected in the no-clearance group were mostly located in the lumen, and had not been internalized or attached to the animal’s tissues.

### 3.3. Nanoparticle Uptake by Oncogenic Flies 

#### 3.3.1. Whole-Abdomen Measurements 

Considering the tentative but not statistically significant uptake of SERS NPs by wildtype Oregon-R and *w^1118^* flies, we next assessed *esg^ts^-Ras^V12^* flies expressing the Ras^V12^ oncogene in all their midgut progenitor cells. Such female adults develop tumors that eventually cover the whole midgut. Within 5 days of induction, the midgut is transformed into a continuous tumor and females typically start dying past 6 days of oncogene induction [20]. Moreover, these flies exhibit normal or even increased defecation rate at 5 days of oncogene induction [20]. Accordingly, the tumorigenic flies were assessed for NP uptake and retention at 2, 4 and 6 days of oncogene induction, at which time points they were fed with SERS NPs for 24 h followed by 3 h of intestinal lumen clearance. The acquired Raman spectra demonstrated low intensities and rich spectral signals for all the experimental conditions (Figure 4a). Moreover, the Raman background signal obtained was low and constant for all 3 time points. The highest signals were collected from 2-day-old flies, followed by the 4-day-old flies. The 6-day-old flies showed no discernible nanoparticle uptake. This result is better identified in Figure 4b, where the nn-LS scores on the SERS nanoparticle reference are shown. Interestingly, the specific NP signal intensity was significantly higher compared to the control at 2 days of oncogene induction. We conclude that tumorigenic flies take up and retain NPs in their dysplastic gut, although less so when closer to their demise, presumably due to appetite loss. The lower retention may be exacerbated due to increased shedding by the tumorous cells, making it harder for NPs to stay attached to the tumorous tissue.

#### 3.3.2. Excised Gut Measurements 

To visualize the nanoparticle distribution within the midgut, we performed an imaging study using excised midguts from *esg^ts^-Ras^V12^* flies expressing the Ras^V12^ oncogene in their midgut progenitors. Tumorigenic flies were collected at 2 days of oncogene induction, and fed with SERS NPs for 24 h followed by 3 h of intestinal lumen clearance. Then, their midguts were dissected, as described in the Section 2, and placed on a glass microscope slide for imaging and signal acquisition. Following Raman imaging, the pixel-wise nn-LS scores on the nanoparticle reference were used to create an intensity image, superimposed on the photograph of the NP-fed fly midguts (Figure 5a,b) and of the control vehicle-fed fly midguts (Figure 5d,e). Most of the NP-fed fly midguts exhibited strong NP signal spots appearing to emanate from the lumen of the posterior midgut (Figure 5a,b). The microscope photograph of the posterior midgut spots of higher Raman signal had a reddish metallic hue, indicating the presence of nanoparticles (Figure 5(bii)). These spots show a high concentration of specific SERS nanoparticle signal in the characteristic Raman bands of the SERS reporter (Figure 5c), and have higher intensity (and nn-LS) scores than the corresponding controls (Figure 5d,e), the signal bands of which were not NP specific (Figure 5f). 

## 4. Discussion

The present study, aimed to assess the retention of SERS nanoparticles in two tumor-free *Drosophila* strains, *w^1118^* and Oregon-R, as well as in the highly tumorigenic *esg^ts^-Ras^V12^* flies. In non-tumorigenic flies, the aging effect was put into consideration, by examining the retention of NPs in 4- and 30-day-old Oregon-R and *w^1118^* flies. For the tumorigenic flies, the effect of the dysplastic gut development was assessed in 2-, 4-, and 6-day-old individuals. For the *w^1118^* and Oregon-R strains Raman signal measurements were taken without and upon gut food clearance, while the *esg^ts^-Ras^V12^* flies were tested only upon luminal content clearance. The intention of measuring the Raman signal in non-tumorigenic flies was to assess SERS-NP uptake in young and healthy versus old flies that exhibit gut leakiness. We used the Smurf assay developed in the fruit fly for measuring gut leakiness upon aging and midgut dysplasia [22,26]. But despite their high incidence of Smurf phenotype, old flies did not exhibit significant NP uptake upon luminal content clearance, producing for the most part intrinsic Raman spectra. This indicated that the NPs did not permeate the lumen or attach to the old midgut epithelium. However, NP-fed flies showed a trend of slightly higher Raman values, compared to the control vehicle-fed flies, indicative of low-level NP acquisition, which was not statistically significant of a difference in the population mean measurements. Our attention was drawn to a single old fly measurement, shown in Figure 3b, that showed a clear characteristic SERS signal of high value, which could be due to NP accumulation in an old fly’s tumorigenic midgut, since up to ~10% of wild-type flies develop spontaneous midgut tumors at that age [27].

Therefore, the assessment of SERS nanoparticle retention in non-tumorigenic young and old flies, especially after gut clearance, does not seem to provide information regarding the permeability state of the *Drosophila* midgut, which was revealed to change through aging via the Smurf assay. This might be due to the size of the NPs used (approximately 100 nm), which might be too big to escape through the *Drosophila*’s leaky gut, so they were excreted from the fly’s body prohibiting significant NP signal detection.

Accordingly, we moved on to examining Raman spectra of tumorigenic flies 2, 4 and 6 days upon the Ras^V12^ oncogene induction in their midgut progenitor cells after 3 h of food clearance by starvation. NP signal detection was significantly higher in the 2-day-old flies compared to that of vehicle-fed control flies, with a *p*-value of 0.036. A similar trend was seen in the 4-day-old flies, but the differences were not statistically significant; while 6-day-old flies showed no discernible NP uptake. The decrease in NP-specific Raman signal from 2 to 4 and 6 days of oncogene induction suggests flies might be losing their appetite closer to their prospective day of death. For example, tumor-secreted factors induce fly cachexia and appetite loss [28]. In that case, the absence of Raman signal in 6-day-old flies maybe be because flies are less able to feed themselves due to tumor-caused anorexia. An alternative mechanism may be that NPs do not attach and are eliminated more quickly as the gut becomes more dysplastic, due to the higher shedding rate of tumorous cells [20]. 

Since the Raman measurements of 2-day-old tumorigenic flies indicated a high retention of the SERS NPs, a further examination was performed to verify that the NPs are actually retained in the oncogenic midguts of these flies. Area scans were performed to dissected midguts upon gut food clearance and then the scans were further analyzed with non-negative least squares analysis, to reveal the spatial distribution of the distinctive SERS NP spectra in each midgut. The results of the signal analysis indicated specific spots of the excised midguts containing distinctive Raman signal, implying the presence of NPs on those areas. Remarkably, when the signal analysis results were superimposed onto the real scanned areas of the microscope slide, the regions containing NPs corresponded to posterior midgut regions. The presence of NPs in oncogenic *Drosophila* midgut was confirmed under the optical microscope. SERS NP accumulation could be seen in the fly’s midgut lumen by their distinctive reddish-metal color. NPs seemed to be attached to the apical side of the tumorigenic epithelial layer facing the lumen, without any indication of absorption. This agrees with the capability of SERS NPs to accumulate in cancer tissues, although NP retention in tumors may also relate to the enhanced barrier permeability noticed in the tumorous vs. healthy epithelia [29]. The probability of NPs being present in the fly’s midgut lumen as a result of constipation is unlikely, since previous studies have shown that the induction of the Ras oncogene in the midgut progenitors does not hinder the fly’s defecation rate [20].

## 5. Conclusions

In the present study, SERS gold nanoparticles are shown to be retained in the oncogenic midgut of the tumorigenic Ras^V12^ expressing *Drosophila*, indicating a propensity to accumulate on tumors. NPs did not to accumulate in healthy flies significantly, despite gut permeation revealed via the Smurf assay, likely due to their size (>100 nm). Moreover, our findings reinforce the usefulness of *Drosophila* combined with SERS, for the study of intestinal dysplasia and tumorigenesis. Future studies using the *Drosophila* SERS model may exploit localized midgut tumors to gain information on the way NPs attach on the intestinal tumor cells. Such studies may introduce functionalized SERS NPs with bio-ligands on their surface targeting specific intestinal tumor epitopes. However, to capture the gut leakiness that develops in the aging *Drosophila*, SERS NPs would need to be redesigned in smaller sizes, since the SERS NPs we used did not seem to penetrate the midgut epithelium. In future studies, SERS NPs within a smaller size range may achieve better tissue penetration and the ability to detect intestinal barrier disruption. Further insight might also be gained via time course measurements of NP retention upon a brief period of NP administration to tumorigenic flies, and by using alternative, less prone to tumorous cell shedding, genetic models of midgut tumorigenesis.

## Figures and Tables

**Figure 1 cells-13-01344-f001:**
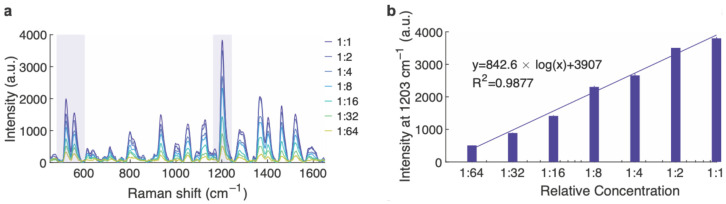
SERS intensity at various nanoparticle relative concentrations. (**a**) The SERS nanoparticles demonstrate a characteristic spectrum that diminishes as they are serially diluted. The characteristic peaks appearing at 1203, 520 and 557 cm^−1^ are highlighted. (**b**) A regression model was applied to the intensity of the 1203 cm^−1^ peak, showing a logarithmic response of the nanoparticles as a function of concentration.

**Figure 2 cells-13-01344-f002:**
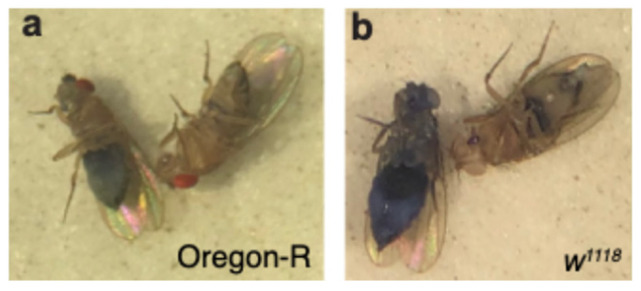
Examples of (**a**) OR and (**b**) *w^1118^* Smurf females with light diffused blue color (**left**) and non-Smurf (**right**).

**Figure 3 cells-13-01344-f003:**
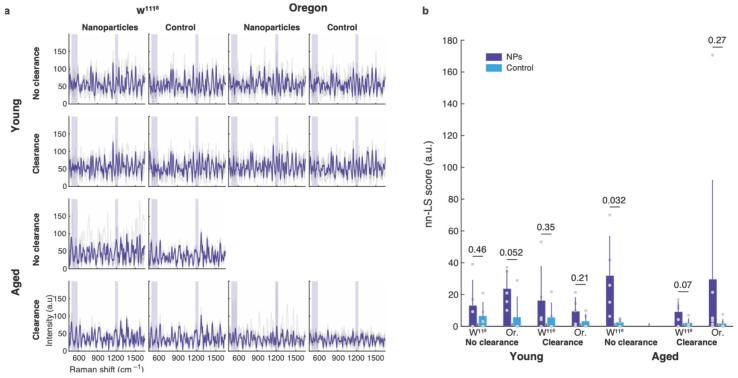
SERS NP uptake by young and old females. (**a**) Raman spectra from young and old flies, *w^1118^* and OR, with and without clearance. The spectra demonstrate many Raman peaks intrinsic to the fly body. Spectra of individual flies are shown in grey and the group average in blue. The shaded bands indicate the areas of the peaks specific to the SERS NPs. (**b**) Regression (nn-LS) scores indicate moderate SERS signals in the flies fed with SERS NPs, with no statistically significant differences, as indicated by the *p*-values. Dots are individual fly measurements, the bars show the mean, and the whiskers the standard deviation. For each condition, 5 to 7 flies were scanned.

**Figure 4 cells-13-01344-f004:**
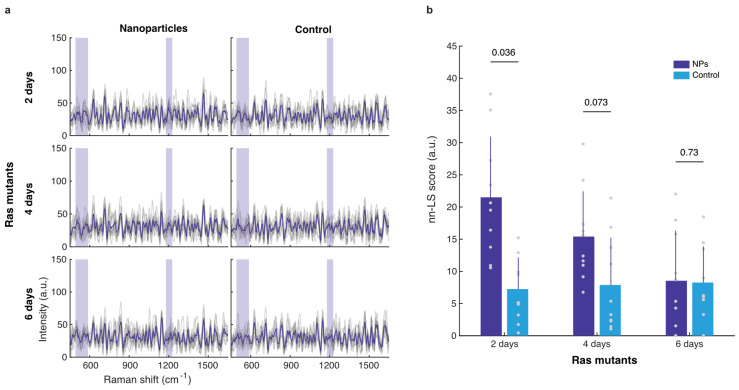
SERS NP uptake by mutant flies. (**a**) Raman spectra for esg^ts^-Ras^V12^ flies on day 2, 4 and 6 of induction of Ras^V12^ oncogene in their midgut progenitors. Spectra of individual flies are shown in grey and the group average in blue. The shaded bands indicate the areas of the peaks specific to the SERS NPs. (**b**) Mean nn-LS scores and *p*-values as calculated for the different groups, show a small but statistically significant uptake for the 2-day condition, which decreases with disease progression. For each experimental condition, 10 flies were scanned.

**Figure 5 cells-13-01344-f005:**
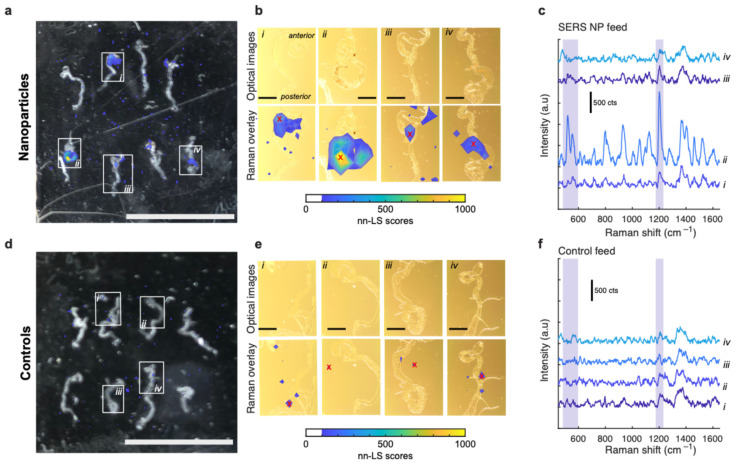
Raman maps of excised fly midguts. (**a**) Photograph of excised guts for NP-fed *esg^ts^-Ras^V12^* flies. The superimposed colormap represents the nn-LS scores obtained from the different areas. Scale bar: 10 mm. (**b**) Detailed view from the sample areas indicated by boxes in (**a**). The optical microscopy images are shown with and without the colormap superimposed, for comparison. The areas with high nn-LS scores in their posterior midgut region display a reddish metallic texture, indicative of the presence of SERS NPs, particularly evident in (**bii**). Scale bar: 1 mm. (**c**) Representative Raman spectra corresponding to the pixels from panel (**b**) indicated with red crosses (×). Area (**bii**) features an exceptionally bright signal, whereas other areas display the characteristic peaks at lower intensities. (**d**–**f**) Similar to panels (**a**–**c**) but from control flies not fed with nanoparticles. Regression signals and Raman spectra have no indication of SERS nanoparticles, as expected.

**Table 1 cells-13-01344-t001:** Smurf and non-Smurf 4-day-old (young) and 30-day-old (aged) OR and *w^1118^* females after Bromophenol Blue feeding. The differences are significant per Fisher’s exact test.

Strain	OR				*w^1118^*			
Condition	Young		Aged		Young		Aged	
	*n*	%	*n*	%	*n*	%	*n*	%
Total	63	100.0	17	100.0	42	100.0	24	100.0
Dead	2	3.2	1	5.9	2	4.8	3	12.5
Non-Smurf	58	92.1	9	52.9	34	81.0	11	45.8
Smurf	3	4.8	7	41.2	6	14.3	10	41.7
*p*-value	0.0004				0.0124			

## Data Availability

Data available upon reasonable request.

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
