# Peer review of "Nanoparticle Uptake in the Aging and Oncogenic Drosophila Midgut Measured with Surface-Enhanced Raman Spectroscopy"

_cells, 2024, doi:10.3390/cells13161344_

Round 1
Reviewer 1 Report
Comments and Suggestions for Authors
The manuscript by Christou et al, entitled "Nanoparticle uptake in the aging and oncogenic Drosophila midgut measured with surface enhanced Raman spectroscopy", examines the uptake of surface-enhanced Raman scattering nanoparticles (SERS NPs) as a marker for the detection and imaging of intestinal tumours to propose an alternative to colonoscopy. They first synthesised nanoparticles and then assessed their Raman signal after feeding control flies of different ages and tumorigenic flies. The authors show that the nanoparticles are retained only in the oncogenic midgut, suggesting a propensity to accumulate in tumorigenic tissues. The overall conclusion reached by the authors is interesting and well supported by their analysis, but I have a few comments to make.
General comments
1)- It would have been accurate to correlate the size and location of the tumour in the gut with the analysis in figure 4, as it is easy to express GFP in addition to RasV12.
2)- The authors stated that the decrease in NP accumulation in the gut at 6 days is due to the flies eating less. It would be worth addressing this point as it is important to further validate their framework for tumour growth.
Minor comments
1)- Figure 2 table 1, it would be useful to indicate the number of different experiments carried out.
2)- The manuscript contains several errors in the referencing of the figures (line 228, 258, 259, 262, etc.).
3)- Figures 4C and f are missing a legend.
Reviewer 2 Report
Comments and Suggestions for Authors
This is an interesting study which reports the uptake of surface-enhanced Raman scattering nanoparticles (SERS NPs) as a marker for intestinal tumor detection in Drosophila for developing alternative diagnostic methods to detect malignancies.
Although the project is well conceived, I think that deserves further investigation concerning excised guts. The authors indicate that SERS gold nanoparticles are retained in the oncogenic midgut of the tumorigenic RasV12 expressing Drosophila, but I understood that this could be true only after 2 days. What happens to the gut of older sick flies? They should perform gut analysis at 4 and 6 days and explain why the NPs are no longer retained. Could it just be related to lack of appetite?
I have minor changes to suggest Figure 2 Picture are too dark, letters are out of focus and smurf flies should have same position in the picture (left or right) Figure 3 and 4; I Can’t see the shaded bands Figure 4 b and 4 e; every single figure in each panel should have the same intensities as each other.The writing is concise and clear and discussion should take in account new results, but is exhaustive and complete.
I don’t recommend the publication of this manuscript in the present form.
Reviewer 3 Report
Comments and Suggestions for Authors
While the article is well-written and the topic appears to be interesting, I believe the following concerns warrant either a major revision or rejection from publication in Cells
- Regarding the Smurf test (Table 1), there are several points to consider:
a) The authors' use of only female flies is not explained in the text. This choice should be justified, as gender differences could potentially influence the results.
b) The unequal number of test flies between groups is a significant concern. The difference is indeed substantial: W1118 shows a 43% difference, while OR flies show a 73% difference.
c) To ensure the relevance and reliability of the results, this test should be repeated with an equal number of specimens in each group. The current imbalance could potentially skew the results and lead to inaccurate conclusions.
- Regarding the overall study:
a) The authors do not present any statistically significant results. They mention that SERS NPs are used in cancer studies in different animal models, but fail to draw strong conclusions from their own data.
b) This article appears to be more of a short report demonstrating that fruit flies can be used to study the molecular mechanisms of SERS NPs. As a short report, it may be suitable for publication.
c) However, the results presented may not be substantial enough for publication as a full article. The lack of significant findings and the methodological issues mentioned above (particularly in the Smurf test) limit its impact as a comprehensive research paper.
In conclusion, while this work has merit as a preliminary study or short report, it would require more robust data, equal sample sizes, and statistically significant results to be considered for publication as a full research article.
Round 2
Reviewer 2 Report
Comments and Suggestions for Authors
I would thank the authors for accepting my suggestions. I am satisfied and have nothing else to ask for.
Author Response
- I would thank the authors for accepting my suggestions. I am satisfied and have nothing else to ask for.
We thank the reviewer for the time they committed evaluating our manuscript.
Reviewer 3 Report
Comments and Suggestions for Authors
Dear Authors, I have carefully reviewed the revised version of your manuscript. I am pleased to note that the paper has significantly improved following the initial review process. Your efforts to address the previous comments and suggestions are evident and have enhanced the overall quality of the work. However, I believe there are still areas that could benefit from further clarification, particularly in the statistical methodology section.
Elaboration of the statistical methodology description
I suggest a more detailed discussion of the applied statistical methods, including:
Precise justification for the selection of specific statistical tests
Explanation of the assumptions underlying the chosen analyses
Clarification of differences in sample size
Please provide:
Clear indication of the reasons for these differences
Discussion of the potential impact of variable sample size on the results
Expanding on these aspects will significantly strengthen the methodological side of the article and allow readers to fully understand the research process. Furthermore, it will increase the possibility of replication of the study by other scientists. Your attention to these points will further elevate the quality of your already improved manuscript.
